# User Perceptions and Needs Analysis of a Virtual Coach for Active and Healthy Ageing—An International Qualitative Study

**DOI:** 10.3390/ijerph191610341

**Published:** 2022-08-19

**Authors:** Johanna Möller, Roberta Bevilacqua, Ryan Browne, Takamitsu Shinada, Sébastien Dacunha, Cecilia Palmier, Vera Stara, Elvira Maranesi, Arianna Margaritini, Eiko Takano, Izumi Kondo, Shuichiro Watanabe, Michael Ahmadi, Rainer Wieching, Toshimi Ogawa

**Affiliations:** 1Diocesan Caritas Association for the Archdiocese of Cologne, 50668 Cologne, Germany; 2IRCCS INRCA, Scientific Direction, 60124 Ancona, Italy; 3Smart-Aging Research Center, Tohoku University, Sendai 980-8575, Miyagi, Japan; 4Faculté de Médecine, Université de Paris, Maladie d’Alzheimer, 75015 Paris, France; 5Service de Gériatrie 1&2, AP-HP, Hôpital Broca, 75013 Paris, France; 6Department of Rehabilitation Medicine, National Center for Geriatrics and Gerontology, Obu 474-8511, Aichi, Japan; 7Institute of Gerontology, J. F. Oberlin University, 3758, Tokiwa-machi, Machida-shi 194-0294, Tokyo, Japan; 8Institute for New Media and Information Systems, University Siegen, Kohlbettstr. 15, 57072 Siegen, Germany

**Keywords:** active and healthy ageing, virtual coach, older adults, digitalization

## Abstract

Virtual coaching systems show great potential for meeting the challenges of demographic change. However, the proportion of older users in the field of digital technologies is far behind that of younger people. As part of the e-VITA project, semi-structured interviews were conducted in Japan, France, Italy and Germany with 58 people aged 65 and over, and the content was analyzed with the aim of obtaining information about how older adults organize their everyday lives, also with regard to the COVID-19 pandemic, how they deal with their health, what role digital technologies play in the lives of the interviewees and why they oppose progressive digitization. Second, the survey asked why the older adults oppose a virtual coach, which is to be developed in the e-VITA project to support older adults in healthy and active aging, and what barriers they see in a possible implementation. It was found that older respondents lead active, varied lives and that the COVID-19 pandemic contributed to the increased use of digital solutions. In addition, respondents were consciously addressing their own health. With regard to a virtual coach, barriers were seen primarily in the area of data security and sharing. It can be concluded from this that heterogeneity among older user groups should be taken into account when developing virtual coaches. In addition, aspects of data security and data protection should be presented in a clearly understandable and transparent manner.

## 1. Introduction

The ageing of the population is one of the most pressing societal and policy issues in the 21st century. The number of Europeans aged 80+ is set to rise from 4.9% currently to 13%, and the group of seniors aged 65+ will represent roughly 28% of Europeans in 2070, with consequent challenges in terms of demands for healthcare, the needs of an ageing population and the sustainability of the workforce [1]. Unlike the EU, Japan has the highest proportion of older adults in the world, 28% in 2018 [2]. Ageing is not only an immediate personal issue but also a salient factor in crucial public policies, such as pensions, health and long-term care.

This scenario requires conceptually new prevention, care and smart-living solutions to support human-based care. In this context, smart technology offers considerable potential. Indeed, technological progress in recent years has provided numerous hardware and software solutions to facilitate active and healthy ageing (AHA) as a way of optimizing opportunities for health, participation and security in order to enhance quality of life as people age. Empowering older adults and their community through user-centered innovation, as well as service delivery, secures independent living and prevention, thus assisting them to live longer in their own homes with the possibility of acting independently and participating actively in society.

Although the number of users of smart technologies is rising steadily, not only among younger people but also among older people, the over-65s still make up by far the smallest group among Internet users, despite the advantages mentioned and the progress of digitization [3,4,5]. While the proportion of Internet users in Germany among the over-70s was still around 29% in 2014, 52% of older adults were using the Internet in 2020. Nevertheless, a large gap is still evident in comparison with the younger generation, as the proportion of Internet users between the ages of 14 and 49 is now almost 100% [3]. A similar pattern can be seen in France. There, 100% of 18 to 24-year-olds use the Internet, whereas the proportion is 81% for 60–69 years old and only 58% for those over 70 [4]. Looking at daily Internet users in Italy in the year 2019, it is also noticeable that there is a large gap between older adults aged 65 and over and younger people. While the 25–34 age group makes up the largest group of daily Internet users (85%), those over 65 constitute the smallest share of users (29%) [5]. The use of the Internet is also steadily expanding among older people in Japan. According to [6] the Ministry of Internal Affairs and Communications’ Survey of Telecommunications Usage Trends (2018), 75.7% of people in their 60s, 53.6% of people in their 70s and 23.4% of people aged 80 and older use the Internet, and the usage rates for all of these age groups increased in 2016 compared to 2010. In this environment of widespread Internet use, the ability to understand the Internet correctly and make informed choices on one’s own will be useful in terms of maintaining health. Reasons for non-use can be manifold: In a British study, [7] investigated the most frequently cited reasons for non-use (people who have never used the Internet) and no longer use (ex-users, people who have used the Internet in the past and no longer do so). Important reasons for not using the Internet were a lack of interest (50%), as well as a lack of access and high cost and lack of skills (10%). In the ex-users group, lack of access and high cost were the main reasons cited for no longer using the Internet.

In this paper, we are providing intercultural insights from a project that actively aims to evoke change by collaborating with older adults from Europe and Japan. Research activities in the “e-VITA” project deal with smart-living solutions for elderly care and active ageing. The main purpose of e-VITA is to improve the well-being of older adults in Europe and Japan by promoting active and healthy ageing and preventing cognitive, physical, emotional and social decline. The partnership will jointly develop and connect innovative smart-living solutions that address individual, as well as cultural, aspects and factors of AHA and overall well-being through AI, smart data analysis and tailored ICT-based interventions and real-life coaching.

In our desire to understand the needs, wishes and attitudes of older adults towards the advancing digitalization, as well as towards the planned coach, we followed a qualitative and explorative approach conducting semi-structured interviews with 58 older adults from Europe and Japan. It is based on data collected during the first six months of the e-VITA project. It aims to collect and analyze end-user requirements for the development of an advanced intercultural virtual coaching system that improves the well-being of older adults in Europe and Japan. Designing a system that aspires to be closely interwoven with the domestic life and everyday activities of older adults at a cross-national level requires a profound understanding of end-user requirements.

## 2. Materials and Methods

Our approach to the fieldwork was a semi-structured interview study. The data presented in this paper were collected during the early project phase between March and April 2021. Overall, 28 participants from three European countries (Italy, France, Germany), as well as 30 participants from Japan, participated (Table 1).

The interviews aimed to receive preliminary information about the daily life (before/after COVID-19) of the potential users of the coaching system, how they deal with their own health and their experiences, as well as attitudes towards different technologies and digitalization. In addition, information about initial expectations regarding the e-VITA coach and speech interaction should be gathered and possible initial barriers regarding the implementation of the virtual coach identified. The interviews were conducted using a semi-structured guide based on the structure of [8], which was initially written in English and then translated for each national language. Before the interview, the interviewer briefly introduced him/herself and thanked the respondent for his/her willingness to take part in the interview. The e-VITA project was then explained, and the goals of the interviews were described. Interviewers were careful not to convey any negative stereotypes towards older adults to mitigate the risk of distorting the interview [9]. It was explained that the interview would last about an hour, that it would be recorded and then transcribed and that no conclusions could be drawn about natural persons. After the introduction, as well as after the interview, there was room for open questions from the participants.

The guide consisted of a total of five topic sections, each of which was introduced by an open-ended narrative prompt (Appendix B Table A1). Topic block 1 “Everyday Life” served as an introduction to the actual interview and pursued the goal of obtaining knowledge about the everyday structure and activities of the interviewees. Topic block 2 “Health” served to determine the importance of health in the interviewees’ everyday lives and obtain information about the personal handling of health. In addition, information about health literacy was to be determined. From this, initial indications of possible areas of application for the virtual coach can already be derived. In topic block 3 “Technology/Digitalization”, information was collected on which digital and technological devices the respondents already use and their basic attitudes towards technology and digitalization. Topic block 4 “e-VITA Coach” introduced the transition to project-related interview content. Here, statements regarding the acceptance of a possible virtual coach were determined, which specific usage scenarios would be conceivable for the respondents and which barriers and opportunities they see in the use of the coach. Topic block 5 “speech interaction” specifically targeted the ideas of language interaction with the coach and what expectations the potential users have of it. Before the start of the interview (Japan) or after conducting the interview, demographic data of the interviewee was collected.

Due to the ongoing SARS-CoV-2 pandemic, the interviews were conducted digitally via different platforms in the European setting which means that the respondents were interviewed in their homes. Before the start of the interview (Japan) or after conducting the interview, the demographic data of the interviewee was collected. In Japan, in addition to digital platform interviews, interviews were conducted in-person owing to the relaxed restrictions during the pandemic.

### 2.1. Inclusion Criteria

According to the e-VITA project activities, the subjects were recruited, mainly within the research organizations carrying out the study, their network, human resource offices and local companies. Basically, people aged 65 and over who had an email address were included. In addition, the subjects had to live mainly independently in their own homes. This means that the test persons could manage the essential activities of daily living (ADL) without major (professional) help from third parties. Persons younger than 65 years or without a valid email address were excluded, as well as persons who needed (professional) help to cope with their essential ADL and thus could not lead a completely independent life. In addition, a pilot test was conducted with a 58-year-old female from the circle of acquaintances of one of the project partners.

### 2.2. Sample Description

A total of 58 older adults (33 women) aged 64–89 ($\overset{-}{X}$ = 73.66) years were interviewed (based on an interesting person’s profile, one person was included in France with an age of 64 years). This resulted in audio material totaling around 72 h. Table 1 shows the characteristics of the interviewees broken down by country and overall.

### 2.3. Data Analysis

The interviews were analyzed using a content analysis based on [10] via the software MAXQDA. For this purpose, a uniform coding system was first created deductively on the basis of the guideline explained above, in which the main categories and subcategories were each explained in terms of content. This coding system formed the basis for the evaluations in all countries (Table 2). The partners were free to add new inductive categories based on their interviews. Each country evaluated their own interviews, and the results were then discussed and compiled in the whole plenary session as part of a triangulation strategy.

## 3. Results

The individual main categories and subcategories broken down by country are presented in Appendix A.

### 3.1. Everyday Life

Respondents in the different countries reported an active life with many different activities. In this thematic area, the activities could be categorized into family and social contacts, voluntary work and hobbies. In France (FR), Germany (GER) and Japan (JP), volunteering was a major part of the respondents’ lives. For example, a German interviewee reported on her voluntary work: “Yes, I am a counsellor at the children’s and young people’s helpline ‘Nummer gegen Kummer.’That is supposed to be nine hours a month in shifts of two or three hours, so that is almost every week, and then I am also a reading mentor at school” (DE_06). Furthermore, contact with the family in everyday life was reported in all European countries and Tohoku. For example, a German respondent describes what contact with his grandchild means to him: “I say, dealing with, with grandchildren, that is not just a grandpa or grandma task, that is a joy of life. Because you see how the children develop” (DE_01). In addition, social contacts outside the family also play a major role, as was reported especially in FR and GER. For example, a German respondent describes his social network as follows (DE_07): “So, I am, yes, as I said, married, am present in a network of about 20, 30 friends, some of them eternal friends since childhood”. A variety of different hobbies were also reported. These included sports (GER, FR, IT, JP) or cultural activities, such as attending concerts or museums (GER, FR). Household activities were part of daily life in all countries. The same is true for religious activities except for France. In this respect, an Italian respondent summarizes his day as follows: “IT_03: “I go out for a walk, I go shopping, I meet some friends, I go to the church.”

The everyday life of the interviewees was not unaffected by the COVID-19 pandemic, which has been presenting the world with new challenges for almost two years now. In all countries, it is described that many activities can no longer be carried out; a Japanese respondent describes it as follows JP_007: “The whole year has been a pain, as I keep saying… I’ve lost a year of my life”. A German interviewee (DE_08) describes in this regard, for example, that his voluntary activities have now ceased “A bit that has changed, I no longer do these reading mentors at the moment and the ‘Bahnhofsmission’ [Christian charity at railway stations] is closed. I do go there regularly to look after things, but it’s just not busy anymore, so I see that I’m mostly alone.” Contact with family is also limited in these times. One Italian respondent describes how their own activities have been restricted and how contact with their daughter and grandson has been reduced (IT_01): “Several things have changed as compared to the past. In the last year life has changed radically. I go out little or nothing due to restrictions, I see little my friends, my daughter and my grandson.”.

But there were also positive aspects to the COVID-19 pandemic. For example, new activities have been begun as a result of the pandemic situation. For example, a German respondent reports that he now helps out as a new activity in a COVID-19 test station (DE_08): “What has happened now is that I have been doing Corona rapid tests at [specific place] since January. (…) (19) I really enjoy that.” Furthermore, digitalization (see Section 3.3) became more part of the respondents’ everyday life and enabled new opportunities, as a French respondent reports that she now bakes cakes with her granddaughter via Skype (FR_03): “With my granddaughter we made cakes by Skype”, and a German respondent explains in this respect that she was able to develop her digital competence in the last year DE_02: “Well, of course I am learning, of course I have also learned a lot in, in the last year in the topic of video conferencing (…)”.

### 3.2. Health

When asked what role one’s own health plays in everyday life, the respondents in the European countries and in Japan show that they consciously deal with their health in everyday life and want to actively influence it. A Japanese respondent describes in this context, that he sees health as his basis for life (JP_008): I do take care of myself, and that’s the basis of my life.” Important aspects of the respondents’ own health are the preservation of physical functioning and independence (FR_04: “Being able to be active and independent”). Social interaction and the avoidance of loneliness were also mentioned in this context, as one German respondent (DE_05) reports: “(…) so quite a lot of contacts, which are extremely, extremely important for ME as healthy ageing becomes the topic at all and I can only imagine it with difficulty, but I know that there are also other people that one can also be really happy with few social contacts.” Fears and worries, in turn, mainly concern the loss of physical functioning and independence, as well as the occurrence of specific illnesses. In FR and JP, the loss of cognitive functioning and the occurrence of Alzheimer’s disease were specifically addressed. For example, a Japanese respondent described as his greatest fear losing himself through a dementia-related personality change (JP_011): I think I’m most afraid of losing track of who I am and what I am.” One particular concern mentioned by a German respondent (DE_07) was the shortage of doctors in rural areas: “In the meantime, it is a bit more difficult because our doctor only takes a certain quota per day, yes, so even at the family doctor’s, you have to reckon with waiting times of six to seven weeks for examinations as a private patient. On top of that, of course, we hardly have any doctors left in rural areas, yes. They all retired a few years ago, yes.”

Regarding the procurement of information on the topic of health, three major categories could be identified: Internet, medical institutions (such as doctor, public health nurse) and other media (such as TV or magazines), as described by an Italian respondent: (IT_04): “I listen to the news every day, I am very aware of health news”. In addition, it was reported that friends and family are consulted for advice or that previous knowledge from education and work is used (GER). However, some respondents also avoid too much information about health topics (FR), as described by a French respondent (FR_12): “In addition, I am afraid that there is too much information and I am afraid of having all the diseases that I read about”. The health-related area of life is also not unaffected by the COVID-19 pandemic. For example, the interviewees report that they have partly become more sensitive to their health and pay more attention to it, as well as the fear of being infected with the coronavirus itself (IT_05): “Before Covid I was less concerned about health, now for me the fear of Covid is a constant concern”. Furthermore, it was reported that important preventive and check-up examinations had to be postponed (GER), that the ongoing pandemic situation leads to more stress (FR) and that alcohol consumption decreased or even increased due to the pandemic (Tohoku, Japan).

### 3.3. Technology in Everyday Life

In general, the respondents had a positive attitude towards new technologies and digital change; an Italian respondent describes it, for example, as an opportunity (IT_01): “I believe that technology can help people in many ways”. However, the participants also have critical aspects in mind, as DE_06 describes: “Yes, it has two sides, I think. I think it makes a lot of things very convenient. What bothers me right now, because I’ve studied it, is that you can no longer see through what actually happens with a swipe, with a click, with a tick somewhere on a form or something.”. In addition, respondents across all countries use a wide range of different devices, from smartphones to tablets with specific apps, desktop PCs, wearable devices or even smart home technologies, as a German respondent (DE_07) describes: “I have a smart home in part of our house here, so roller shutter control was one example, or heating control. I would say I have a relatively high affinity for technology.” In addition to the number of different devices that the respondents already used, different usage scenarios could be identified. These were divided into three categories, everyday life, health and social activities. In the area of everyday life, technical devices were mainly used to play games, read, watch videos or to obtain information quickly. For example, a German respondent (DE_04) reports that she always quickly searches for information that interests her at the moment. “(…) so you always have a portable encyclopaedia with you, right? (laughs) Then you just think, man, now I’ve read a book somehow by (unv.) or something like that. Where is (unv.) or something, and then I look it up and then I know that and then I have this link and then I keep it.” In the area of health, technical devices were also used in the lives of older adults. Thus, it is reported that vital signs are checked independently and that contact is made with doctors to discuss it or that technical devices are used for sports, as a French respondent reports (FR_01): “I also did yoga with the Internet and YouTube.” In the category of social activities, respondents described how they use technical devices to establish and maintain contact with friends and family, like IT_03: “luckily I have mobile phones, so I can call my sons and grandchildren”, or DE_03: “And I (need?) that so, as far as communication is concerned and I can send a picture. Take a picture, send a picture, so right now, I have grandchildren, quite a few, and I communicate with them in this way, that I simply ask, do you have time, will you come over, or they say, I’ll come over then and then”. The interviewees therefore see the advantages of digitalization above all in the area of social contacts, especially that these are easier to maintain and establish, against the background of the COVID-19 pandemic. (JP_009): I use it the most because I have free phone calls. I send out postcards to friends I don’t see due to corona, saying I called because I want to hear their voices”. In GER, for example, video conferencing was seen as a great asset in this context: “(…) so I would NEVER have thought of holding a face-to-face event online cleverly. And I am SO excited about it (…)” (DE_05). Furthermore, it was described how technical devices “make life easier”, for example that it is easier to obtain certain information or that certain devices can support activities that would otherwise no longer be possible. For example, DE_03 reports that “(…) I can’t write by hand very well anymore, so I write everything on the PC.”

In addition to the advantages that, from the respondents’ point of view, are associated with technical devices and digitalization, a number of different disadvantages were also reported. Among them, above all aspects was that in the area of social interactions and bad usage patterns, as well as negative effects on health. However, topics such as concrete barriers to use, phishing, advertising and environmental aspects were also addressed. Negative aspects on social interactions played a role especially in FR, IT and JP (TOHOKU, NCGG). For example, a Japanese respondent describes how he feels disturbed by the constant accessibility on his smartphone (JP_r01): “I had been in touch with [volunteer colleague] several times via e-mail and phone, but when I missed his e-mail, I’d get a message asking, ‘Where are you?”. Another major aspect, also reported in FR and JP (Tohoku), was the negative impact on health, such as specifically on the eyes, memory or posture. In this context, a French participant described how the use of new technologies causes one’s own memory skills to fade into the background (FR_04): “I think it doesn’t make you work on your memory any more”. Concrete barriers to use in JP were, on the one hand, the fear of phishing, the design of displays (mostly perceived as too small) and the lack of compatibility with the Japanese written form and native words (JP_r01): “Everything is written in horizontal letters and using foreign terms, right?”. In GER, features perceived as useless discouraged participants from using different technical devices and the fear of not keeping up with technological progress was a frequently mentioned aspect here. For example, a German participant reported that people who are already 10 years older than her have already missed the boat (DE_04) “(139) But I think it is important. Well, I sometimes think that those (.) who are now ten years older than I am somehow have simply MISSED the connection.”. In FR, personalized advertising and the impact of electronic devices on the environment were specifically seen as negative. Regarding the latter, a French respondent describes the consequences of increased electricity consumption on climate change, (FR_06: “The cost on the climate is a very high cost in terms of electricity consumption”).

### 3.4. Attitude towards a Virtual Coach

When asked what the participants understood by the term “virtual”, most associated it with not being confronted with a “real” person (FR_09: “that everything is through a phone or a screen is all. There is no human in front of it”). With regard to the appearance of a possible “virtual coach” as planned in e-VITA, various ideas could be identified. These include human-like, animal-like, abstract forms or (concrete) robots. Regarding the first, for example, a French respondent reports that she would prefer a human-like shape to a more functional shape (FR_09: “I would prefer a robot that has a bit of a humanoid shape rather than a robot like an automatic hoover”) and a Japanese respondent reports that the virtual coach could look like his pet (JP_005: “It would be nice if we could work together in the garden, like a pet”). However, some respondents also mentioned already existing devices or specific robots like NAO or PEPPER, as a German respondent explains (DE_07): “So Pepper is already quite good. Well, he has a few characteristics that are positive, he has a friendly face, can smile and so on (…)”. Regarding the human-like or animal-like appearance of a virtual coach, respondents from GER and JP said that the virtual coach should not look too realistic, that they had the feeling that they would be fooled by a too-realistic appearance: “But do I have to imagine a pretty blonde or a cat or a penguin there? Nah. That makes me feel like I’m being screwed” (DE_01).

In the further course of the interview, the respondents were asked about possible use scenarios in the areas of social life, everyday life and health. Regarding the first, the possibility of the coach establishing contact with friends or family members was mainly described (GER, FR, JP). For example, according to a French respondent, the coach could suggest people who have not been spoken to for a while (FR_09 “On the other hand, it could suggest to the person to think about calling someone (…)”). Furthermore, a German respondent (DE_02) reports that the virtual coach could also establish contact with strangers who have the same interests or with people who have a different mother tongue, in order to expand one’s own language skills: “(87) the coach would then ask me, yes, do you possibly want to improve your French skills, because you might want to go on holiday to France. And I would suggest this online discussion group, where native speakers of the same age from France participate, with whom you can now speak and communicate and you speak in French and the others speak in German”. On the other hand, it is reported how the virtual coach itself could serve as a social contact and provide the user with company or emotional support. In this regard, for example, an Italian respondent reports that the coach could calm her down (IT_02: “It should calm me down and give me some emotional support”), and Japanese respondents report that the coach should be a part of life and develop with the external circumstances (JP_r02: “listen to music with me, or read my favourite books for me, or share some part of my life with me” or JP_r03 “the robots should grow and develop as they respond to their environment… I would like to live together with it for a long time”).

In the area of everyday life, mainly reminder functions, information provision and the suggestion of certain activities by the virtual coach were described. Providing information means that the virtual coach can answer specific questions from the user and provide further information on certain topics. In this regard, a German respondent reported that the virtual coach could, for example, play out contents of films if the person concerned is visually impaired (DE_04: “Yes, for example, reading out current information, that is, somehow, when it becomes more difficult to see, and the other could also be played out in such a way, so that perhaps, there are now films where it is described what is happening or so”). In addition, the respondents reported that the virtual coach could give suggestions for activities in the vicinity, and a French respondent commented as followsL (FR_10: “He could suggest entertainment or activities, theatre, concert, music”). Furthermore, it could also be possible to book tickets directly via the coach, as a German respondent (DE_05) suggested: “you can then book the theatre ticket, that is, yes, that is such a very concrete support or motivation”.

Concrete usage scenarios could also be identified in the area of health. These could be divided into the categories of diagnosis, emergency situations, reminder functions and health advice. With regard to the diagnosis category, the respondents describe how the virtual coach could check vital signs and, if necessary, warn if values are in a pathological range or, in addition, directly make diagnoses. A German interviewee (DE_07) describes how the virtual coach could identify abnormal values and then forward them directly to a specialist, “That is, it would be good that a watch, for example, or such an assistance system says: Watch out, (it’s time?). I measured your blood pressure, it’s over 140. It’s high time you did something. Or your urine, (urine level?) is so negative now, something’s wrong. Please see a specialist. And it would be best to make an appointment (unv.) for the consultation, then that would be optimal”. Another example is the idea of a Japanese respondent who suggests that the virtual coach could analyze sleep and then give suggestions for the optimal duration and time to get up (JP_001): “I am curious about how deep my sleep is, so if it could show me a diagram, and I would know when I can wake up comfortably, it would be very useful”. The usefulness in emergency situations mainly referred to the fact that the virtual coach could recognize emergency situations independently and then notify the emergency doctor or family members (FR_07): “(256) Instead of picking up the phone and calling 15 we would have a dashboard and we would have to press a button and automatically [inaudible] that would ask you what you have etc., that could take your blood pressure from a distance etc”. In the area of health, the idea of a reminder function was also frequently expressed, e.g., to take medication, to carry out sporting activities or to have preventive check-ups. For example, a German respondent (DE_08) described: “A step further would be to be reminded of certain appointments, or to take tablets regularly, to drink regularly, and perhaps also to exercise. So that means that this digital coach also keeps me on my toes a bit or reminds me, here, you haven’t done any gymnastics today. Move at least a little bit. At the moment my wife does that when I’m lazy”. With regard to the COVID-19 pandemic, the interviewees saw possible use scenarios primarily in the fact that the virtual coach could inform them about the current incidence and the regulations associated with it or remind them of the hygiene measures they should pay attention to.

However, despite the versatile use scenarios, the respondents also formulated concrete concerns and barriers to use. These included fears of dependency and loss of control, being controlled by the device and ethical and data protection concerns. However, concerns about funding were also expressed. Regarding the feeling of losing control or becoming controlled and dependent, a French respondent describes how this should be counteracted (FR_10: “it should be flexible and docile, so that I can go back to it if I have an unexpected problem. It shouldn’t have too much effect on my life. It should not manage my life. I should feel that I am in control of my life, even if he helps me and offers me advice, I should still be in control. If we ever slip up, he shouldn’t make me feel guilty. There should be safeguards at that level”). In this regard, it was also important to the respondents that they know all the functions beforehand when purchasing a virtual coach, that they have enough time to familiarize themselves with the device and know how to react if problems occur or that a human contact person is still available for support. Across the countries, fears regarding data protection and the fear of the possible misuse of personal data were mentioned. For example, a German respondent (DE_04) reported that she would definitely like to have control over the data processing: “But I would already want to know how I can operate it and what the basic functions are, what data it collects and where this data goes. And I would also like to be able to control the data and delete it if necessary. So that would be important to me”. A French respondent reports how she would feel if health data were simply passed on (FR_11: “I’d be embarrassed if he used the health data, I gave him without asking me and without telling me what it would be used for”).

### 3.5. Speech Interaction

The idea of having a normal conversation with a technical device led to a mixed picture among the respondents. On the one hand, participants found it difficult to imagine this situation because they assumed that it would be limited by technical boundaries. Furthermore, a virtual coach would lack emotional skills that make up a conversation between two people. For example, a Japanese respondent says (JP_003) “I think that no matter how much progress is made, AI can’t have emotions… No matter how advanced the machine is, I think it’s just a mechanical robot that tells us what it knows”. On the other hand, respondents see it as a way to combat loneliness, as an Italian respondent describes (IT_01: “I would like to have a conversation with the robot. Even though I know it would not be a conversation with a real person, it would still keep me company”).

Regarding the nature of the language, the respondents hoped that the language should not sound too “robotic”, that the voice could be adapted or that the voice adapts to circumstances on its own. For example, a French respondent reports that the virtual coach should adjust its voice output if it registers that the user does not understand the statement correctly (FR_08: “if he understands that I am repeating the same thing several times, and either he needs to raise his voice or he needs to have a slower phrasing”). In addition, the interviewees partly explain that the virtual coach should not always adapt to their views, but that he should also be able to lead a discussion sometimes. For example, a Japanese respondent describes it as follows: (JP_010) “I feel like I want to argue with a robot. “No, I don’t like that. I’d like to try annoying the robot. It would be nice to have someone who replies, “What are you talking about?”… It’s good for brain activity.”

## 4. Discussion

The aim of this study was to identify needs, perceptions and possible barriers for the development of an advanced intercultural virtual coaching system that improves the well-being of older adults in Europe and Japan. To this end, semi-structured interviews were conducted in Japan (n = 30), Italy (n = 8), France (n = 12) and Germany (n = 8) as part of a qualitative preliminary study of the e-VITA project.

The results from Topic 1, everyday life, shows, that across all countries, it was observed that the participants were living an active and social life. The results hence do not back the stereotype of the inactive and isolated older adult [9]. Above all, social relationships and the availability of social contacts played an essential role in the lives of the people interviewed. At this point, however, first, cultural differences become clear. When it comes to religion and spirituality, in GER and IT, it was mainly the community and social aspect that motivated people to participate in church activities. In Japan, this topic is treated more individually. The reason for this could be that the Christian religion in Western countries is strongly institutionalized and thus has an important influence on coexistence in society. In Japan, on the other hand, there has never been a comparable authority such as the Catholic Church in Europe (Yanagawa, 1991). In the case of Topic 2, one’s own health, participants in all settings were already pretty aware of their health-related issues. In Italy, several participants stated that the pandemic made them more aware of their health state. A mixed-methods study by [11] quantitatively and qualitatively surveyed Phillipines on their reactions to the spread of COVID-19, which also showed that health consciousness has become a focus of the respondents due to the pandemic. However, in this regard, it must be noted that there are different forms of health consciousness, and, in the present study, as well as in the study of [11], it could be disease-centered health consciousness, which according to [12] is only a reaction to a specific disease and is usually not sustained over a long period of time. Further studies should examine whether the COVID-19 pandemic has had a significant impact on population health consciousness and whether this may even extend beyond disease-specific health consciousness. For health-related advice, the participants in all settings saw the potential benefits of a virtual coach for a reminder or recommendation features. We also observed country-related differences regarding health-related information consumption. For instance, interview partners from Italy stated that they mainly received their information from the TV, while Japanese and German ones receive them from the Internet. In order to be able to classify information found on the Internet, respondents in Germany in particular drew on previous medical knowledge from their profession or education. At this point, technology should support the user in filtering out information from the Internet, as not everyone will have prior medical knowledge.

Topic 3, technology usage in everyday life, all participants were regular PC and/or smartphone users. During the pandemic, they increasingly learned how to use technologies to stay in contact with relatives and acquaintances, indicating a general openness to technology as a measure for social interactions. Some participants were also vivid users of self-tracking devices or apps. A study [13] examined technology-use behaviors during the COVID-19 pandemic among 2909 older adults and supports the conclusion of the present analysis. In their analysis, ICT use increased for more than 50% of respondents during the COVID-19 pandemic. When asked what the person imagined by the word “virtual”, clear differences between the respondents could be identified. Some of the participants had problems describing what they imagined, while others were even able to name certain robots. Hence, the future technological devices need to take different technical competencies that reflects the diversity of older adults into account. However, this could also be due to the fact that the formulation of the question could have been too abstract for some participants and that a different formulation would have led to specific ideas about the term “virtual” for these respondents as well. As some participants faced frustration in setting up devices or software in the past, they asked for a contact person in case of technical difficulties (help desk).

With regard to the concrete practical use of the virtual coach, the respondents could imagine that all of those social contacts could be established and maintained through the system. Opinions differed on the idea that the virtual coach itself could be a social ally or even an emotional support. For example, in Japan and Italy, it was quite conceivable that the coach would also respond emotionally to the users’ needs or be a life companion. In Germany (but also partly in Italy, too), on the other hand, this function was seen rather critically and declared as “fake”. It is also important to mention that concerns about the coach being not trustworthy enough and too commanding was frequently stated in all countries. In this vein, participants want to preserve a sense of control over the device. Furthermore, a vital worry, mentioned across all settings, was privacy issues. In this respect, it is of great relevance that users can clearly determine which of their personal data is to be used and that the use and transfer of the data are transparent and easy to understand. Moreover, participants across all contexts were worried about potential financial barriers. Thus, regarding a market solution, one has to manage the trade-offs of satisfying diverse needs and expectations while creating low-cost solutions.

## 5. Strength and Limitation

The strength of the present study is the multicultural approach given by the four participating countries. This allows for a multifaceted survey and, despite the specific inclusion criteria, a heterogeneous sample, which is important in qualitative research to be able to map the variance of the population [14]. Despite this strength, gender disparities in the sample size, could be seen as bias and significant limitations that do not allow for the generalization of results: on the side of the European countries, mainly women were interviewed and, in addition, on the Japanese side, more than half of the interviewees were female. However, existing studies show that there is a difference in the use of modern technologies between diverse genders [15]. This should be taken into account in subsequent qualitative studies in order to gain deeper insights into gender-specific differences in use. In the German sample, moreover, all of the respondents worked regularly with digital technologies and actively expressed their desire to participate in the study, and all (except for one person) performed a volunteer work. This suggests that the respondents may have been particularly motivated and open to new technologies in principle. Further studies should therefore analyze the attitudes and impressions of people who do not use new technologies at all or who hardly use them or are critical of them. Nevertheless, in the present study, there were also partial difficulties in understanding on the side of the interviewees. For example, the interviewees sometimes found it difficult to imagine the voice interaction between a virtual coach and the user. The reason for this could be that the simple mention of a “virtual coach” without a more detailed explanation of the function was not sufficient for the participants, making it difficult for the interviewees to transfer an interpersonal interaction to an interaction with a digital technology.

## 6. Conclusions

In summary, virtual coaching systems and future technologies that aim to improve the quality of life of older adults should take into account the pronounced heterogeneity of the target group. This also implies that care should be taken not to reproduce negative stereotypes with respect to ageing. Older people should be individually met where they stand with their digital skills. Potential users with existing knowledge should be taken seriously and not be underchallenged, and users with no prior knowledge should not be overchallenged. This also concerns the provision of (medical) information. Since the Internet is used to gather information on health-related topics, the virtual coaches should support the user in filtering out serious information from the masses. Not everyone can access prior knowledge, as was the case with some of the respondents in this study. In addition, a virtual coach should support the maintenance of existing social contacts and assist in establishing new contacts while taking into consideration different social needs. Digital solutions that focus on religious needs should take into account older adults’ individual understanding of religion and spirituality, as well as cultural differences. In order for a virtual coach to be used by the older target group, it should be ensured that the user has control over the device itself and over the use of personal data. In addition, all aspects concerning the storage and protection of sensitive data should be presented transparently and be easy for the user to understand. In addition, when developing virtual coaches for active and healthy ageing, the influence of the external appearance of the technology should not be underestimated. Due to the different preferences that were mentioned in our interviews, this topic should be investigated in more detail in further studies, also with regard to cultural differences. This also affects the user interface, which should be adapted to the cultural and individual needs of the users (e.g., the reading direction of the font, font size, icon size, etc.). Further studies should also survey the attitudes of people who do not use digital devices and who use them only to a limited extent. Last but not least, if technologies that promote active and healthy ageing make it to the market, they should be accessible to all social strata and not only be available to privileged people who have the financial means to do so. Otherwise, this could increase health inequalities. Within this study, we found empirical evidence for the characteristics that a virtual coach should have: the coach needs to promote autonomy, encouragement and empowerment instead of patronizing and instructing older adults. This has direct implications for the proactivity of the voice coach, as well as the level of control older adults expect to have over the voice coach. Moreover, given the diversity of the target group enrolled, it becomes apparent that there is a need for personalization with regard to cultural and personal preferences (e.g., religion). Consequently, interfaces (e.g., tone of voice) are subject to appropriation by individuals. Any coaching platform must provide reasonable defaults but also offer enough modularity to allow customized experiences. It is notable that older adults were also aware of their own ageing process and expected the coach to adapt to their changing needs. Keeping these relations in mind while designing functions helps to provide a consistent experience to the end users and improve the acceptance of the technology.

## Figures and Tables

**Table 1 ijerph-19-10341-t001:** Interview and participant characteristics.

Country	Data Collection	Participants
Germany	Interviews via Zoom April 2021 Duration of audio recording: 01:05:44–01:28:08 h (∑ = 10:26:05 h)	n = 8 (4 female) Age: 66–87 Years ($\overset{-}{X}$ = 71.63 Years)
France	Interviews via Skype March 2021 Duration of audio recording: 01:11:58–01:55:24 h (∑ = 17:50:07 h)	n = 12 (7 female) Age: 63–87 Years ($\overset{-}{X}$ = 74.17 Years)
Italy	Interviews via phone, in-person March 2021 Duration of audio recording: 01:07:35–01:23:07 h (∑ = 9:57:27 h)	n = 8 (7 female) Age: 68–88 Years ($\overset{-}{X}$ = 80.75 Years)
Japan	Interviews via Zoom, in-person April, May 2021 Duration of audio recording Tohoku: 0:43:21–2:34:51 h average: 01:30:47 h (∑ = 25:43:20 h)	Tohoku: n = 17 (8 female) Age: 67–83 years ($\overset{-}{X}$ = 73 Years)
	NCGG: 00:12:46–00:41:56 h average: 25:29:24 h (∑ = 2:07:27 h)	NCGG: n = 5 (1 female) Age: 65–73 years ($\overset{-}{X}$ = 69.8 Years)
	IGOU: 00:15:19–00:56:46 h average: 45:26:30 h (∑ = 6:03:32 h)	IGOU: n = 8 (8 female) Age: 61–89 years $\overset{-}{X}$ = 71.5 Years

**Table 2 ijerph-19-10341-t002:** Cross-national basis of the category system for the analysis of the interviews in MAXQDA.

Main Category	Subcategory 1. Level	Subcategory 2. Level
Everyday activities		
	During the week	
	Weekend	
	References	
	Under COVID-19	
Health		
	Importance in everyday life	
		Conscious confrontation
		No conscious confrontation
	Worries and fears	
	References	
	Information gathering	
	Influence of COVID-19	
Technologies in everyday life		
	Attitudes towards technologies/digitalization	
		Rather positive
		Rather negative/skeptical
	Specifically used devices	
	Use scenarios	
		Health-related
		Everyday life
		Social life
e-VITA Coach		
	First unspecific ideas	
	Appearance of the coach	
	Possible use scenarios	
		Under COVID-19
		Social life
		Everyday life
		Health-related
	Concerns/barriers	
	Requirements for use	
Speech interaction		
	First idea on interaction	
	References	
	Barriers	

## Data Availability

Not applicable.

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
