# Peer review of "User Perceptions and Needs Analysis of a Virtual Coach for Active and Healthy Ageing—An International Qualitative Study"

_ijerph, 2022, doi:10.3390/ijerph191610341_

Round 1
Reviewer 1 Report
The research is very interesting one but it needs to narrow down hugely in order to focus a specific objective. At present it is written just a report of a research project.
Author Response
We would like to thank the reviewers for the all the comments and suggestions for improving the manuscript. We have tried our best to address all of them, without missing the scope and contents of our article.
In particular, in order to make the contents more systematic, we have added a Table in Annex 1, at the end of the manuscript, that summarizes all the results we have collected. The dissertation in the text is necessary for understanding the improvements from the end-users perspective around the most important themes to design the eVITA virtual coach and to link them together in order to create a clear integrated explanation of the future design choices. This process allows us to emerge with some over-arching themes that can be helpful in tying the individual pieces of data together as already expressed by relevant theoretical publication on qualitative analysis methodology and reporting style (Rubin & Rubin, 1995, Kvale, S. (1996). InterViews: An introduction to qualitative research interviewing. London: Sage Publications; Lindlof, T. R. (1995). Qualitative communication research methods. Thousand Oaks, CA: Sage Publications; Rubin, H. & Rubin, I. (1995). Qualitative interviewing: The art of hearing data. Thousand Oaks, CA: Sage; Silverman, D. (2006). Interpreting qualitative data: Methods for analysing talk, text and interaction (3rd ed.). London: Sage Publications.).

Reviewer 2 Report
The title of the article is not related to the contents presented.
Participants have given suggestions about a virtual coach they did not know. These opinions can help to shape the properties and solutions to be provided by the virtual coach.
Information about the sample configuration is scarce and common selection criteria are not known.
There is a lack of information on: their level of digital competences; socio-economic level of the selected persons, education, etc. . The researchers acknowledge the limitations of the sample only in the limitations section.
The strength of the study lies in the qualitative methodology. The researchers need to improve the systematisation of the results obtained. The analysis should be done by categories and subcategories (responses in different countries, etc.).
The use of a table would help to differentiate the responses by country, religion, gender, age, education, etc.
The conclusions should be related to the characteristics that a virtual coach should have.
Author Response
Dear reviewers,
We appreciate the careful and thoughtful comments made by you and we are happy to expand our manuscript to address your concerns. We hope that you, will find our comments and revisions responsive to the issues raised.
R1: The title of the article is not related to the contents presented.
A1: The title has been changed to be more aligned with the contents of the paper.
R2: Participants have given suggestions about a virtual coach they did not know. These opinions can help to shape the properties and solutions to be provided by the virtual coach.
A2: We thank the reviewer for this comment. The work that we have reported here is the preliminary step we have ran for designing the eVITA system, as correctly addressed by the reviewer. We have used all the observations from the users to shape the virtual coach characteristics that is actually under experimentation.
R3: Information about the sample configuration is scarce and common selection criteria are not known. There is a lack of information on: their level of digital competences; socio-economic level of the selected persons, education, etc…The researchers acknowledge the limitations of the sample only in the limitations section.
A3: We thank the reviewer for this comment and we have added 146-148 and we have added some consideration in the limitations section in lines 526-529 and in the conclusion in lines 574-586. The description of the sample includes now is reported in Table 1. Unfortunately, as highlighted in the limitation section, there was not possible to collect information on educational level and literacy.
R4: The strength of the study lies in the qualitative methodology. The researchers need to improve the systematisation of the results obtained. The analysis should be done by categories and subcategories (responses in different countries, etc.). The use of a table would help to differentiate the responses by country, religion, gender, age, education, etc.
A4: Following the reviewer suggestions, we have added an Annex A, that summarizes the results on the basis of the following subcategories. The table of the Annex A is available at the end of the manuscript and it is introduced in lines 174-175. Moreover, we thought that the description of the results should be favoured by the dissertation around most important themes, instead of reporting the description of the results for subcategories, as the comparative analysis allow us to refine concepts, and to link them together in order to create a clear description or explanation of the main theme under study. In this way, the concepts and themes we have found should be put together to build an integrated explanation, which should then be interpreted in the light of the literature and the theories presented in your theoretical framework. This process allows us to emerge with some over-arching themes that can be helpful in tying the individual pieces of data together as already expressed by relevant theoretical publication on qualitative analysis (Rubin & Rubin, 1995, Kvale, S. (1996). InterViews: An introduction to qualitative research interviewing. London: Sage Publications; Lindlof, T. R. (1995). Qualitative communication research methods. Thousand Oaks, CA: Sage Publications; Rubin, H. & Rubin, I. (1995). Qualitative interviewing: The art of hearing data. Thousand Oaks, CA: Sage; Silverman, D. (2006). Interpreting qualitative data: Methods for analysing talk, text and interaction (3rd ed.). London: Sage Publications.).
R5: The conclusions should be related to the characteristics that a virtual coach should have.
A5: The conclusion has been expanded in lines 574-586.

Round 2
Reviewer 2 Report
The authors have given an adequate response to the improvement proposals